# Multicenter Evaluation of the Cepheid Xpert^®^ HBV Viral Load Test

**DOI:** 10.3390/diagnostics11020297

**Published:** 2021-02-12

**Authors:** Fabbio Marcuccilli, Stephane Chevaliez, Thomas Muller, Luna Colagrossi, Giulia Abbondanza, Kurt Beyser, Mélanie Wlassow, Valérie Ortonne, Carlo Federico Perno, Marco Ciotti

**Affiliations:** 1Laboratory of Virology, Virology Unit, Polyclinic Tor Vergata Foundation, Viale Oxford, 81, 00133 Rome, Italy; abbondanzagiulia1@gmail.com (G.A.); marco.ciotti@ptvonline.it (M.C.); 2National Reference Center for Viral Hepatitis B, C and Delta, Department of Virology, Hôpital Henri Mondor, Université Paris-Est, 94010 Créteil, France; stephane.chevaliez@aphp.fr (S.C.); melanie_wlassow@aphp.fr (M.W.); valerie_ortonne@aphp.fr (V.O.); 3Medizinisches Versorgungszentrum Labor München Zentrum GbR, SYNLAB MVZ, 92637 Weiden, Germany; thomas.mueller@synlab.com (T.M.); kurt.beyser@synlab.com (K.B.); 4Area of Multimodal Medicine and Laboratory, Children Hospital Bambino Gesù, 00165 Rome, Italy; luna_colagrossi@yahoo.it (L.C.); cf.perno@uniroma2.it (C.F.P.)

**Keywords:** Xpert^®^ HBV viral load assay, CAP/CTM HBV test, v.2, HBV DNA quantification

## Abstract

Accurate measurement of the hepatitis B virus (HBV) DNA is important for the management of patients with chronic HBV infection. Here, the performance of the Xpert^®^ HBV Viral Load test (Xpert HBV Viral Load) versus the Roche COBAS^®^ Ampliprep/COBAS^®^ TaqMan^®^ system (CAP/CTM HBV) HBV test v2.0 was evaluated. From September 2017 to December 2017, a total of 876 prospectively collected or archived serum or EDTA plasma specimens from subjects chronically infected with HBV were tested using the Xpert HBV Viral Load and the CAP/CTM HBV v2.0 assays. Of the 876 specimens tested, 560 were within the quantitative range of both assays. The agreement between the two methods was 90.0%. No difference in plasma or serum samples was observed. Deming regression analysis showed a good correlation of the Xpert HBV Viral Load assay with the CAP/CTM HBV v2.0 assay. The Bland–Altman analysis showed a good agreement between the results of the Xpert HBV Viral Load assay and the CAP/CTM HBV assay, with a mean difference (±1.96 standard deviation) of 0.0091 ± 0.3852 Log IU/mL. Comparing the two assays, only nineteen specimens (2.1%) had a difference greater than 1.96 times the standard deviation. The Xpert^®^ HBV Viral Load test is suitable for monitoring patients with HBV infection and is useful in diagnostic settings.

## 1. Introduction

Chronic hepatitis B virus (HBV) infection is a global health problem with significant morbidity and mortality, being one of the most common causes of cirrhosis and hepatocellular carcinoma [1]. It is estimated that about 257 million people are chronically infected by HBV around the world, with the highest prevalence in the African and Western Pacific regions [2]. Accurate measurement of HBV DNA levels in blood is essential to diagnose HBV infection, establish the prognosis of HBV-related liver disease, and guide the treatment decision to treat and monitor the virological response to antiviral treatment and the emergence of resistance [3]. In recent years, several real-time PCR- or TMA-based assays have been introduced in routine diagnostics to monitor HBV chronic patients [4,5,6,7]. Such assays are characterized by high sensitivity, a wide dynamic range, and genotype inclusivity. However, currently available HBV DNA assays are generally designed for batch testing of multiple specimens within a run. 

The Xpert^®^ HBV Viral Load test (Cepheid, Sunnyvale, CA, USA; CE-IVD (in vitro diagnostic medical devices)) is a new a new real-time PCR assay for HBV quantification run on the automated GeneXpert^®^ Systems that allows continuous loading of specimens with true random access. It provides results in about 1 h and 30 min. This fast turn-around time translates into faster delivery of medical reports available for patients care. 

This study was conducted to evaluate the performance of the Xpert^®^ HBV Viral Load test in clinical specimens collected in patients infected with genotypes frequently encountered in Western Europe.

## 2. Materials and Methods

### 2.1. Study Design

Each clinical specimen was tested with two different HBV DNA quantification assays, including the Xpert^®^ HBV Viral Load test and one widely used commercially available assay, the COBAS^®^ AmpliPrep/COBAS^®^ TaqMan HBV test, version 2.0 (Roche Diagnostics, Indianapolis, IN, USA; COBAS^®^ Ampliprep/COBAS^®^ TaqMan^®^ HBV 2.0).

### 2.2. Specimens Collection

In total, 888 fresh and frozen human samples, including serum (*n* = 216) and plasma (EDTA) samples (*n* = 672) from patients with chronic HBV infection, were collected from September 2017 to December 2017 at three health care sites in Western Europe (Polyclinic Tor Vergata, Rome, Italy; Henri Mondor Hospital, Créteil, France; SYNLAB MVZ Weiden, Germany) and supplemented by two locations in the United States (BioCollections Worldwide, Inc., Miami, FL, USA; Bloodworks Northwest, Seattle, WA, USA). Frozen (≤−70 °C) specimens were tested on both platforms (Xpert HBV Viral Load and CAP/CTM HBV v2.0 assays) from the same freeze–thaw cycle; after thawing, samples were immediately processed. Freshly drawn whole blood specimens were stored at 2–8 °C or 15–30 °C prior to centrifugation and were tested within 24 h of collection.

The study was approved by the Institutional Review Boards (IRBs) or Ethics Committees (ECs) of the centers involved in the study.

### 2.3. HBV DNA Quantification

In the Xpert HBV Viral Load procedure, one milliliter of plasma or serum was transferred into the cartridge containing all reagents needed for sample preparation, nucleic acid extraction, and quantification of PCR products. The dynamic range of quantification is 10 to 1 × 10^9^ IU/mL (1.0 to 9.0 Log IU/mL), with a limit of detection (LOD) of 3.20 IU/mL for plasma and 5.99 IU/mL for serum according to the manufacturer’s product insert. The viral region targeted by the primers and probe is the preC-C (Pre-Core-Core) gene.

In the CAP/CTM procedure, HBV DNA was extracted from 650 µL of plasma or serum by means of the Cobas AmpliPrep automated extractor, according to the manufacturer’s instructions. The Cobas TaqMan 96 analyzer was used for automated real-time PCR amplification and detection of PCR products, according to the manufacturer’s instructions.

The dynamic range of quantification of the CAP/CTM HBV v2.0 assay is 20 to 1.7 × 10^8^ IU/mL (1.3 to 8.2 Log IU/mL), with an LOD of 9.0 IU/mL in plasma and 19.0 IU/mL in serum. The viral region targeted by the primers and probe is the preC-C gene.

In cases of differences in results between the Xpert HBV Viral Load and the CAP/CTM HBV version 2.0 greater than 0.5 Log IU/mL, a new aliquot was tested using the Abbott RealTime HBV Assay (Abbott Molecular, Inc., Des Plaines, IL, USA; hereafter “Abbott”) or the VERIS MDx HBV assay (hereafter “Veris”). The viral region targeted by the primers and probe is the S gene for both assays.

### 2.4. HBV Genotype Determination

The HBV genotype was available for 200 samples, including 44 from genotype A, 42 from genotype B, 41 from genotype C, 33 from genotype D, 39 from genotype E, and 1 from genotype F. Genotypes were determined by each participating site by sequencing of a portion of the S gene followed by phylogenetic analysis.

### 2.5. Statistical Analysis

Descriptive statistics are shown as mean values ± standard deviations (SDs). Relationships between quantitative variables were studied by means of Deming regression. A Bland–Altman plot was also used to highlight the differences between the quantification assays.

## 3. Results

Of the 888 collected specimens, 876 samples (215 sera and 661 plasma) were analyzed. The remaining twelve samples (1.3%) had insufficient volume for testing (*n* = 9) or generated indeterminate results (*n* = 3) with the Xpert HBV Viral Load test. 

Of the 876 samples analyzed, 802 (91.5%) had qualitative results that were concordant with the two platforms (Table 1). The fifteen samples quantified with CAP/CTM HBV version 2.0 but not with the Xpert HBV Viral Load assay had a mean (±SD) HBV DNA level of 573 ± 1850 IU/mL, while the fourteen quantified by the Xpert HBV Viral Load assay but not by CAP/CTM HBV version 2.0 had a mean (±SD) HBV DNA level of 103 ± 230 IU/mL. All specimens, except for five with discrepancies, showed a low level of HBV DNA (<100 IU/mL) (Table 2 and Table 3). Some of the samples were retested with the Abbott Realtime HBV or the VERIS MDx HBV assay and the results are presented in Table 4. Twenty-five samples were detectable but not quantifiable on one platform and not detectable on the other platform. No difference was found between serum and plasma samples.

Of the 876 samples analyzed, 560 (including 200 with known genotypes) fell within the dynamic range of quantification of the two assays. Figure 1 shows the relationships between HBV DNA levels measured with the Xpert HBV Viral Load and the CAP/CTM HBV version 2.0 assays. A strong correlation between the two assays was found (coefficient correlation (*r*), *r* = 0.97; Deming regression equation, *Y* = 1.042*X* – 0.15) (Figure 1A).

As shown by the Bland–Altman plot analysis, a weak bias for the HBV DNA level was observed (bias: +0.0091 ± 0.3852 log UI/mL) (Figure 1B). No major difference of quantification was observed according to the HBV DNA level. In 19 samples, the differences between the assays were greater than 1.96 times the standard deviation, including 10 that were higher and 9 that were lower with the Xpert HBV Viral Load assay. The differences were less than 1.0 log IU/mL in all cases except for 11 of these samples.

## 4. Discussion

Accurate and reproducible HBV DNA level measurements in blood specimens are crucial to diagnose HBV infection, establish the prognosis of HBV-related liver disease, and guide the treatment decision to treat and monitor the virological response to antiviral treatment and the emergence of resistance [3,7,8,9,10,11,12]. In the present study based on a large number of clinical specimens, including specimens with different genotypes found in HBV-infected patients, the new real-time PCR-based Xpert HBV Viral Load assay accurately quantified HBV DNA levels in plasma and serum. Its performance appeared to be comparable to those of an existing assay and platform that is widely used in clinical practice, namely the CAP/CTM HBV version 2.0 assay. We observed only a very modest difference in HBV DNA levels when we compared the Xpert HBV Viral Load assay to the CAP/CTM HBV version 2 assay. This small deviation was independent of the HBV genotype and HBV DNA level and most likely has no implications in clinical practice [13].

The study confirms previous reports where a high correlation between the Xpert HBV Viral Load assay and the CAP/CTM HBV version 2.0 assay was found [14,15,16]. The performance of the Xpert HBV Viral Load assay was also evaluated versus the Abbott HBV assay [16] and the Aptima Quant HBV assay [17]. There was a high correlation between the Xpert HBV Viral Load test and these latter assays [16,17]. 

It is worth noting that the Xpert HBV Viral Load assay also has the potential of being used for point of care molecular testing, as demonstrated by the studies by Gupta et al., Woldemedihn et al., and Jackson et al. [16,18,19], especially in developing countries. 

We added some new data respect to the previously published papers. In this study, we used fresh and stored plasma and serum samples, respectively. The Xpert HBV Viral Load assay performs equally well on both specimen types (serum and plasma) and with either stored or fresh samples.

This study has several limitations. First, the small proportion of tested samples containing genotype F reflects the HBV genotype distribution in the different countries participating. Although our results strongly suggest that the performance of the Xpert HBV Viral Load assay is good for all genotypes, further studies, particularly with HBV-infected patients with rare genotypes (G, H, I, and J), are warranted. Second, hepatitis B virus DNA monitoring of patients receiving antiviral treatment is missing. Thirdly, not all discrepant specimens could be tested using another technique due to an insufficient volume in some of them.

Overall, this is the first study showing that the new real-time PCR-based Xpert HBV Viral Load assay accurately quantifies HBV DNA levels in plasma and serum samples from patients with chronic HBV infection. The Xpert HBV Viral Load assay can, thus, confidently be used to detect and quantify HBV DNA in clinical trials with new anti-HBV drugs currently in development and in clinical practice.

## Figures and Tables

**Figure 1 diagnostics-11-00297-f001:**
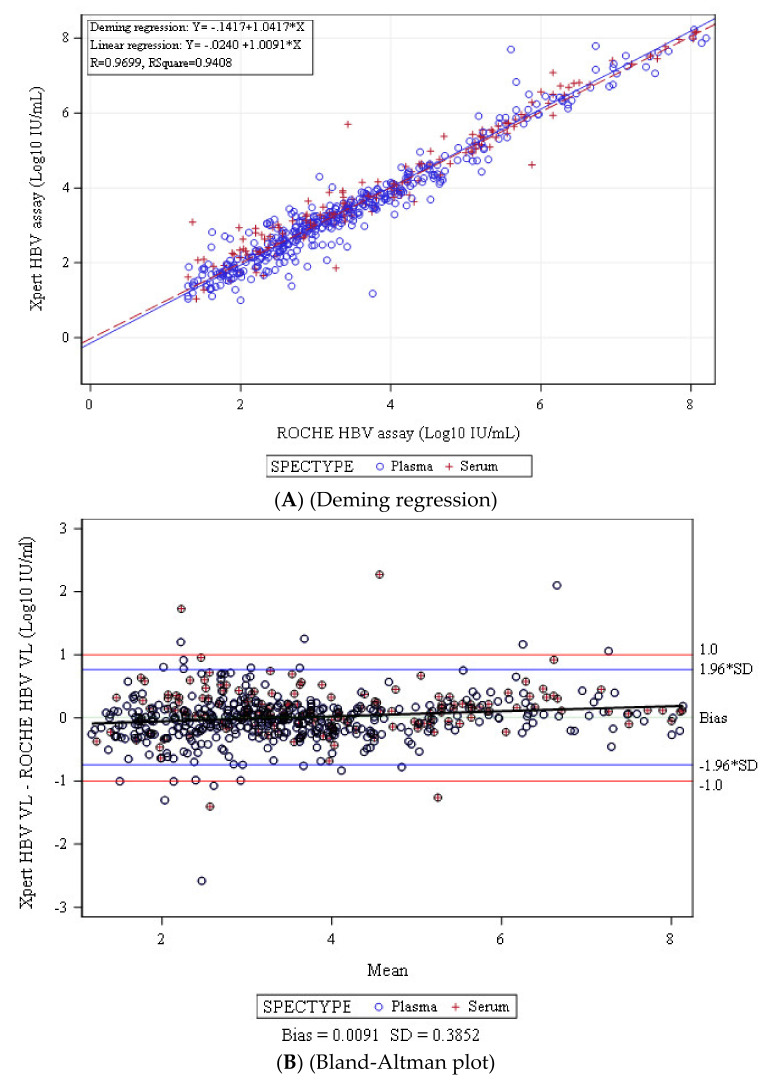
Deming correlation and Bland–Altman-plot analysis of HBV DNA levels measured by Xpert HBV Viral Load and CAP/CTM HBV version 2.0 assays. (**A**) Deming regression of 560 clinical specimens quantified by both assays. (**B**) Bland–Altman plot analysis of 560 samples quantified by both assays. The dotted and dashed lines represent mean differences ± 1.96 SD, respectively.

**Table 1 diagnostics-11-00297-t001:** Qualitative results of Xpert HBV Viral Load and CAP/CTM HBV version 2.0 assays.

	CAP/CTM HBV Version 2.0			
	Virus Detected, ≥20 IU/mL	Virus Detected, <20 IU/mL	Virus Not Detected	No. of Total
Xpert HBV Viral Load				
Virus detected, ≥10 IU/mL	574	13	1	588
Virus detected, <10 IU/mL	6	25	12	43
Virus not detected	9	33	203	245
No. of total	589	71	216	876

**Table 2 diagnostics-11-00297-t002:** Discordant specimen results with Xpert HBV VL quantitative values and Roche HBV assay values <20 IU/mL or not detected.

Specimen ID	Xpert HBV Viral Load (IU/mL)	CAP/CTM HBV Version 2.0 (IU/mL)	Abbott RealTime HBV (IU/mL)
HBV229148	45	<20	QNS for testing *
HBV229167	49	<20	Not tested
HBV229175	47	<20	Not tested
HBV229212	17	<20	Not tested
HBV348175	11	<20	Not tested
HBV364458	13	<20	QNS for testing
HBV364469	20	<20	QNS for testing
HBV380127	17	<20	QNS for testing
HBV380128	38	<20	QNS for testing
HBV380142	13	<20	QNS for testing
HBV380180	72	<20	Not detected
HBV364403	48	<20	<15
HBV229201	166	Not detected	QNS for testing
HBV364442	891	<20	256

* QNS = quantity not sufficient.

**Table 3 diagnostics-11-00297-t003:** Discordant specimen results with Roche HBV quantitative values and Xpert HBV values <10 IU/mL or not detected.

Specimen ID	Xpert HBV Viral Load (IU/mL)	CAP/CTM HBV version 2.0 (IU/mL)	Abbott RealTime HBV or VERIS MDx HBV * (IU/mL)
HBV348237	<10	39	20
HBV348294	<10	43	55
HBV364131	<10	41	<15
HBV364262	<10	87	Not detected
HBV380144	<10	28	QNS for testing
HBV364201	<10	550	Not detected
HBV229156	Not Detected	32	QNS for testing
HBV229223	Not Detected	55	46
HBV364286	Not Detected	55	Not detected
HBV364287	Not Detected	85	Not detected
HBV364409	Not Detected	56	Not detected
HBV364433	Not Detected	31	Not detected
HBV380160	Not Detected	26	<15
HBV364197	Not Detected	224	Not detected
HBV380211	Not Detected	7244	177

* HBV229223 was the only sample tested using the Veris HBV assay; all others were tested using the Abbott HBV assay.

**Table 4 diagnostics-11-00297-t004:** Clinical specimens with discrepancies between Xpert HBV Viral Load, CAP/CTM HBV version 2.0, Abbott RealTime HBV, and VERIS MDx HBV assay results.

Specimen ID	Storage	Xpert HBV VL (Log_10_ IU/mL)	CAP/CTM HBV 2.0 (Log_10_ IU/mL)	Difference (Log IU/mL)	Abbott RealTime HBV/VERIS MDx HBV (Log_10_ IU/mL)
HBV364192		1.18	3.76	−2.58	1.18
HBV348238		1.38	2.68	−1.30	2.59
HBV229152 *		1.63	2.64	−1.01	3.10
HBV380197		1.86	3.27	−1.40	3.52
HBV364256		2.07	3.15	−1.08	2.21
HBV229150 *		2.82	1.62	1.20	2.52
HBV380249		3.09	1.36	1.73	2.98
HBV348195		4.30	3.05	1.25	4.13
HBV348255		4.62	5.88	−1.26	4.88
HBV348220		5.70	3.43	2.27	5.29
HBV364494		6.83	5.67	1.17	6.50
HBV364384		7.70	5.60	2.10	7.48
HBV348100		7.79	6.73	1.06	7.70

Note: * tested using the VERIS HBV assay; all others tested using the Abbott HBV assay.

## Data Availability

Data are available from the corresponding author upon reasonable request.

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
