# Peer review of "Multicenter Evaluation of the Cepheid Xpert® HBV Viral Load Test"

_diagnostics, 2021, doi:10.3390/diagnostics11020297_

Round 1

Reviewer 1 Report

The manuscript entitled "Multicenter evaluation of the Cepheid Xpert HBV Viral Load Test" by Marcuccilli et al., discusses the performance of the Xpert HBV Viral Load test, a new test kit developed by Cepheid to monitor and aid on the diagnostics of patients with chronic HBV infection. New assays for HBV diagnostics are valuable, especially these allowing individual rather than batch diagnostics. Overall the manuscript is well written and scientifically sound, the authors did point the weaknesses of their study in the discussion and introduce this as the first study evaluating the performance of the real-time PCR based assay. My recommendation for the authors is to add a paragraph discussing the overall benefits/importance of this new assay in comparison to others in the market in the introduction. Also, there are a few minor grammar mistakes that should be addressed in the text.

Author Response

The manuscript entitled "Multicenter evaluation of the Cepheid Xpert HBV Viral Load Test" by Marcuccilli et al., discusses the performance of the Xpert HBV Viral Load test, a new test kit developed by Cepheid to monitor and aid on the diagnostics of patients with chronic HBV infection. New assays for HBV diagnostics are valuable, especially these allowing individual rather than batch diagnostics. Overall the manuscript is well written and scientifically sound, the authors did point the weaknesses of their study in the discussion and introduce this as the first study evaluating the performance of the real-time PCR based assay. My recommendation for the authors is to add a paragraph discussing the overall benefits/importance of this new assay in comparison to others in the market in the introduction. Also, there are a few minor grammar mistakes that should be addressed in the text.

Thank you for your comments. We added a paragraph in the Introduction stressing the benefits of this new assay compared to the others already present in the market. We also revised the text for grammar mistakes.

Reviewer 2 Report

Despite the availability of effective vaccines and antivirals based on nucleoside analogs, 350-400 million people worldwide are carriers of the hepatitis B virus, and about 600 thousand die from this disease every year. Accurate determination of the hepatitis B virus (HBV) DNA level allows to make a prognosis of the infection development and assess the risk of complications, such as hepatocellular carcinoma and liver cirrhosis. There is a wide range of PCR based diagnostic test systems for detecting HBV DNA (Roche, Abbott, Chiron, etc.). The authors compared the Xpert® HBV Viral Load test (Cepheid) and the Roche COBAS® AmpliPrep / COBAS® TaqMan HBV Test, version 2.0 by the level of detectable HBV DNA. Altogether 876 frozen human serum and plasma samples were taken for analysis.

The study showed that the results obtained with both test systems coincide in 90% of cases. A strong correlation between the two assays was found (correlation coefficient  = 0.97). This led the authors to conclude that the Xpert® HBV Viral Load test is suitable for monitoring of patients with HBV infection and it is useful in the diagnostic setting. Still, the study has several limitations: only one sample containing genotypes F tested, rare genotypes (G, H, I and J) were not included in the analysis. Nevertheless, the data obtained could be of interest to specialists in the field of infectious diseases and virologists studying hepatitis. However, it should be noted that in 2020 there were already several publications in which the Xpert® HBV Viral Load test and the Roche COBAS® AmpliPrep / COBAS® TaqMan HBV Test were compared with similar results. One of them is also co-authored by Mélanie Wlassowa and Stéphane Chevalieza (Lila Poiteau et al., J of Clinical Virology. 2020, v. 129, https://doi.org/10.1016/j.jcv.2020.104481). The main difference is that not frozen human serum and plasma samples were included in the analysis, but plasma and whole blood collected from dried blood spot. Another study (Ali M. Auzin et al J Med Virol. 2020; 1-5. DOI: 10.1002 / jmv.26392) was carried out on 102 stored plasma samples from patients chronically infected with hepatitis B virus. High correlation was observed between the Roche Cobas HBV Viral Load tests and the Xpert HBV Viral Load Assay (Pearson correlation coefficient r2 = 0.987). The authors do not mention any of these works. Moreover, the most recent references in the bibliography are limited by 2017. I would like the authors to discuss the results obtained in comparison with the results presented in the above-mentioned studies, as well as with other publications on this topic carried out in the last 3 years. In addition, it is necessary to pinpoint what new, previously unknown data were obtained in their study.

Author Response

Despite the availability of effective vaccines and antivirals based on nucleoside analogs, 350-400 million people worldwide are carriers of the hepatitis B virus, and about 600 thousand die from this disease every year. Accurate determination of the hepatitis B virus (HBV) DNA level allows to make a prognosis of the infection development and assess the risk of complications, such as hepatocellular carcinoma and liver cirrhosis. There is a wide range of PCR based diagnostic test systems for detecting HBV DNA (Roche, Abbott, Chiron, etc.). The authors compared the Xpert® HBV Viral Load test (Cepheid) and the Roche COBAS® AmpliPrep / COBAS® TaqMan HBV Test, version 2.0 by the level of detectable HBV DNA. Altogether 876 frozen human serum and plasma samples were taken for analysis.

The study showed that the results obtained with both test systems coincide in 90% of cases. A strong correlation between the two assays was found (correlation coefficient  = 0.97). This led the authors to conclude that the Xpert® HBV Viral Load test is suitable for monitoring of patients with HBV infection and it is useful in the diagnostic setting. Still, the study has several limitations: only one sample containing genotypes F tested, rare genotypes (G, H, I and J) were not included in the analysis. Nevertheless, the data obtained could be of interest to specialists in the field of infectious diseases and virologists studying hepatitis. However, it should be noted that in 2020 there were already several publications in which the Xpert® HBV Viral Load test and the Roche COBAS® AmpliPrep / COBAS® TaqMan HBV Test were compared with similar results. One of them is also co-authored by Mélanie Wlassowa and Stéphane Chevalieza (Lila Poiteau et al., J of Clinical Virology. 2020, v. 129, https://doi.org/10.1016/j.jcv.2020.104481). The main difference is that not frozen human serum and plasma samples were included in the analysis, but plasma and whole blood collected from dried blood spot. Another study (Ali M. Auzin et al J Med Virol. 2020; 1-5. DOI: 10.1002 / jmv.26392) was carried out on 102 stored plasma samples from patients chronically infected with hepatitis B virus. High correlation was observed between the Roche Cobas HBV Viral Load tests and the Xpert HBV Viral Load Assay (Pearson correlation coefficient r2 = 0.987). The authors do not mention any of these works. Moreover, the most recent references in the bibliography are limited by 2017. I would like the authors to discuss the results obtained in comparison with the results presented in the above-mentioned studies, as well as with other publications on this topic carried out in the last 3 years. In addition, it is necessary to pinpoint what new, previously unknown data were obtained in their study.

Thank you for your observations. We added and discussed the two mentioned studies as well as other  studies that evaluated the performance of Xpert HBV Viral Load Assay: Woldemedihn J Virol Methods 289 (2021) 114057; Gupta et al. J Virol Methods vol. 290, April 2021, 114063; Jackson et al J Med Virol 2020 Nov 11. doi: 10.1002/jmv.26662; Abravanel et al Diagn Microbiol Infect Dis. 2020 Feb;96(2):114946. doi: 10.1016/j.diagmicrobio.2019.114946).  Our results confirm the high correlation between Roche COBAS® AmpliPrep / COBAS® TaqMan HBV Test and the Xpert HBV Viral Load Assay as already shown by the two studies performed by Poiteau et al. and Auzin et al.

Gupta et al showed a good correlation between Xpert HBV Viral Load assay and both Roche HBV and Abbott HBV assays. In this study, the most common HBV genotype tested was D. Instead, Woldemedihn et al and Abravanel et al compared Xpert HBV Viral Load vs Abbott HBV and Aptima Quant HBV assays, respectively. There was a high correlation between the Cepheid and Abbott and Hologic assays, respectively. Furthermore, the study by Woldemedihn et al as well as those by Gupta et al and Jackson et al showed that the Xpert HBV Viral Load assay could be particularly useful as point-of-care testing, especially in low-income countries. 

Unfortunately, we could not test samples containing rare HBV genotypes (G, H, I and J) because of the lack of patients chronically infected by such genotypes.

Finally, our study adds some new data respect to the previously published papers.  In this study, we used fresh and stored plasma and serum samples, respectively. We showed that the Xpert HBV Viral Load Assay performs equally well on both specimen types (serum and plasma) either stored or fresh samples.

Round 2

Reviewer 2 Report

A revised version of the manuscript can be published.